# Safety assessment of obinutuzumab: Real-world adverse event analysis based on the FAERS and JADER databases from 2013 to 2025

Xiangpeng Li[1], Haonan Liang[2], Ping Leng[1], Zhanqi Cao[1]*

1 Department of Pharmacy, The Affiliated Hospital of Qingdao University, Qingdao, Shandong, China,
2 Department of Pharmacy, The People's Hospital of Rongcheng, Weihai, Shandong, China

* czq0803@qdu.edu.cn

## Abstract

### Background

Obinutuzumab is the first glycosylated type II anti-CD20 monoclonal antibody for the treatment of lymphocytic leukemia and follicular lymphoma. This research aimed to identify significant and unexpected adverse events (AEs) associated with obinutuzumab by utilizing data from the US Food and Drug Administration's Adverse Event Reporting System (FAERS) and the Japanese Adverse Drug Event Report (JADER) databases, with the intention of providing a reference for the safe and rational clinical use of the drug.

### Research design and methods

The reporting odds ratio (ROR), proportional reporting ratio (PRR), Bayesian confidence propagation neural network (BCPNN), and empirical Bayesian geometric average (EBGM) were employed to analyze the AEs of obinutuzumab using the registration data from the FAERS and JADER databases spanning from 2013 to 2025.

### Results

The study screened 7,868 and 1,584 AE reports related to obinutuzumab from the FAERS and JADER databases, respectively. These AEs involved 198 and 39 risk signals, respectively, and were associated with 16 and 8 system organ classes. In the analysis of the top 30 preferred terms, 19 and 15 risk signals in the FAERS and JADER databases, respectively, were not documented in the drug instruction. Moreover, when obinutuzumab is used for tumor indications, the frequency and signal strength of AEs related to infection and infusion-related reaction (IRR) are higher than those when it is used for non-tumor indications.

**Data availability statement:** All relevant data are within the paper and its Supporting information files.

**Funding:** The author(s) received no specific funding for this work.

## Conclusion

The results of signal mining indicate that more attention should be paid to the risks of obinutuzumab-related AEs. Strengthening clinical medication monitoring is necessary to mitigate the impact of AEs on patients' prognosis and quality of life.

## 1. Background

Obinutuzumab is distinctive third-generation fully humanized type II anti-CD20 monoclonal antibody. It can recognize and eliminate CD20-positive B cells. Anti-CD20 monoclonal antibodies can be categorized into type I and type II based on their mechanism of action and CD20 binding characteristics [1]. Type I monoclonal antibodies, like rituximab, induce significant complement-dependent cytotoxicity. In contrast, type II monoclonal antibodies, such as obinutuzumab, are more effective inducers of antibody-dependent cell-mediated cytotoxicity (ADCC) and direct cell death (DCD), and only weakly trigger complement activation [2]. Compared with rituximab, obinutuzumab lacks a specific immunoglobulin (Ig) G oligosaccharide residue in its Fc region, which enhances its affinity for the FcγRIIIa receptor on immune effector cells [3,4]. In 2013, the US Food and Drug Administration (FDA) approved obinutuzumab for the initial treatment of chronic lymphocytic leukemia (CLL), primary or recurrent refractory follicular lymphoma (FL), as well as subsequent maintenance therapy. Subsequently, since August 2018, obinutuzumab has been used for FL treatment in Japan [5]. Multiple clinical studies have demonstrated that obinutuzumab exhibits good efficacy and safety in treating FL patients and can extend the progression-free survival (PFS) of patients [6–8]. Currently, the guidelines of the National Comprehensive Cancer Network (NCCN) suggest using obinutuzumab in combination with chemotherapy as the first-line treatment for FL. The GALLIUM trial is a randomized, open-label, phase III clinical trial designed to compare the efficacy and safety of obinutuzumab combined with chemotherapy versus rituximab combined with chemotherapy in previously untreated patients with advanced indolent non-Hodgkin's lymphoma (NHL). The results indicated that the PFS in the obinutuzumab plus chemotherapy group was significantly superior to that in the rituximab plus chemotherapy group [9]. However, although the obinutuzumab group demonstrated a significant advantage in PFS, there was no significant difference in overall survival (OS) between the two groups.

In numerous clinical trials, obinutuzumab has demonstrated promising therapeutic and safety advantages. Multiple investigations have indicated that while obinutuzumab exhibits a comparable safety profile, it seems to be more toxic when administered compared to rituximab [6,10,11]. The most severe adverse drug reactions stated in the instructions are IRR, tumor lysis syndrome (TLS), and thrombocytopenia. In a meta-analysis of multiple clinical trials, the overall risk of severe IRR and thrombocytopenia (grade ≥3) caused by obinutuzumab was at least twice that of the rituximab group [7]. Severe thrombocytopenia is most prevalent in the first treatment cycle and is associated with similar risk factors, such as high levels of circulatory

disease, high levels of CD20 expression, splenomegaly, and bone marrow infiltration [12]. Obinutuzumab appears to trigger a more intense cytokine release syndrome, resulting in elevated rates of IRR and thrombocytopenia [1,12]. Thus, with the increasing utilization of obinutuzumab, a number of drug-related AEs necessitate our careful assessment.

The aim of this study was to gather obinutuzumab-related AEs from the FAERS and JADER databases, and to analyze and compare the differences in obinutuzumab-related AEs between the two databases. Given the distinct characteristics of the databases, the recorded AE results would vary [13]. FAERS database received numerous non-severe AEs reported by non-healthcare professionals, whereas JADER database received many severe AEs reported by medical professionals [14]. Moreover, AE reports in FAERS database originated from all over the world, while those in JADER database were restricted to reports within Japan. By analyzing and comparing the collected information on AEs caused by obinutuzumab, potential AE signals were mined to optimize the treatment plans for patients and provide references for the safe and rational use of drugs in clinical practice.

## 2. Data and methods

### 2.1. Data sources

The AE report data employed in this study were sourced from the Japanese JADER database and the American FAERS database spanning from 2013 to 2025. The AE report information from the database starting from the first quarter of 2013 was separately extracted. Subsequently, data from tables such as demographic characteristics and demographic and administrative information (DEMO), adverse drug reaction information (REAC), drug information (DRUG), and others were selected for mining and analysis. The data in the database are primarily submitted as spontaneous reports by medical professionals, consumers, manufacturers, and other parties. In this study, obinutuzumab was chosen as the suspected drug class, and its name was coded using RxNorm. Drug names were standardized using the Medex_UIMA_1.8.3 system to guarantee consistency across all reports. This step is crucial for accurately identifying all reports related to obinutuzumab. Duplicate reports were identified and removed to prevent overestimating AEs. Additionally, the study utilized the Medical Activity Dictionary (MedDRA 26.0) to align the preferred terms (PTs) for AEs associated with obinutuzumab and the corresponding system organ categories (SOCs). Various clinical characteristics, including sex, age, reporting region, reporting identity, reporting time frame, and patient outcomes associated with obinutuzumab-induced AEs, were collected.

### 2.2. Data analysis algorithms

To assess the correlation between drugs and AEs, the indicator employs a four-grid scale to compare the observed frequency ratio between the exposed and unexposed populations. This research utilized a disproportionality approach, which included calculating the proportional reporting ratio (PRR) [15], the reporting odds ratio (ROR) [16], the Bayesian confidence propagation neural network (BCPNN) [17], and the empirical Bayesian geometric mean (EBGM) [18], to detect potential AEs. PRR and ROR are devised based on statistical principles to identify the relationship between drugs and adverse events. BCPNN and EBGM integrate more sophisticated statistical models, enabling them to handle more complex data structures and uncertainties. The rationality of these detection methods in identifying AEs is primarily manifested in their robust statistical foundation, accurate calculation methods, effectiveness in practical situations, and extensive application in the field of pharmacovigilance. They signal AEs promptly when the evaluation criteria are met. The criteria for detecting positive signals were as follows: the number of target event reports of the target drug was ≥ 3, the lower limit of the 95% confidence interval (CI) of ROR was > 1, PRR was > 2, the information component (IC) was > 0, the lower limit of the IC confidence interval (IC-2SD) was > 0, and the lower limit of the modified gamma Poisson shrinker confidence interval (EBGM05) was > 2. The analysis of risk signals was completed using Python version 3.12.0 software. The four methods are based on the assumption that AE reports follow pair-to-pair contingency tables (Table 1). The calculation

**Table 1. Four-cell table of proportional disequilibrium method.**

| Drugs | Target AEs | Other AEs | Total |
|---|---|---|---|
| Target drug | a | b | a+b |
| Other drugs | c | d | c+d |
| Total | a+c | b+d | n=a+b+c+d |

a, number of reports of the target AE associated with obinutuzumab; b, number of reports of other AEs for obinutuzumab; c, number of reports of the target AE for other drugs; d, number of reports of other AEs for other drugs.

formula is presented in S1 Table. Fig 1 depicts a detailed flowchart that outlines the step-by-step procedures of data extraction, cleaning, and analysis.

## 3. Results

### 3.1. Basic characteristics of obinutuzumab-related AEs

In this study, after deduplication in the FAERS database and the JADER database, 7,868 and 1,584 reports with obinutuzumab as the main suspected drug were retrieved, respectively. Analyzing the annual reporting ratios, it was observed that the number of annual reports in the FAERS database has been increasing year by year since 2013, reaching a peak of 1,543 cases in 2024. The JADER database started reporting in 2018, and the year with the highest number of annual

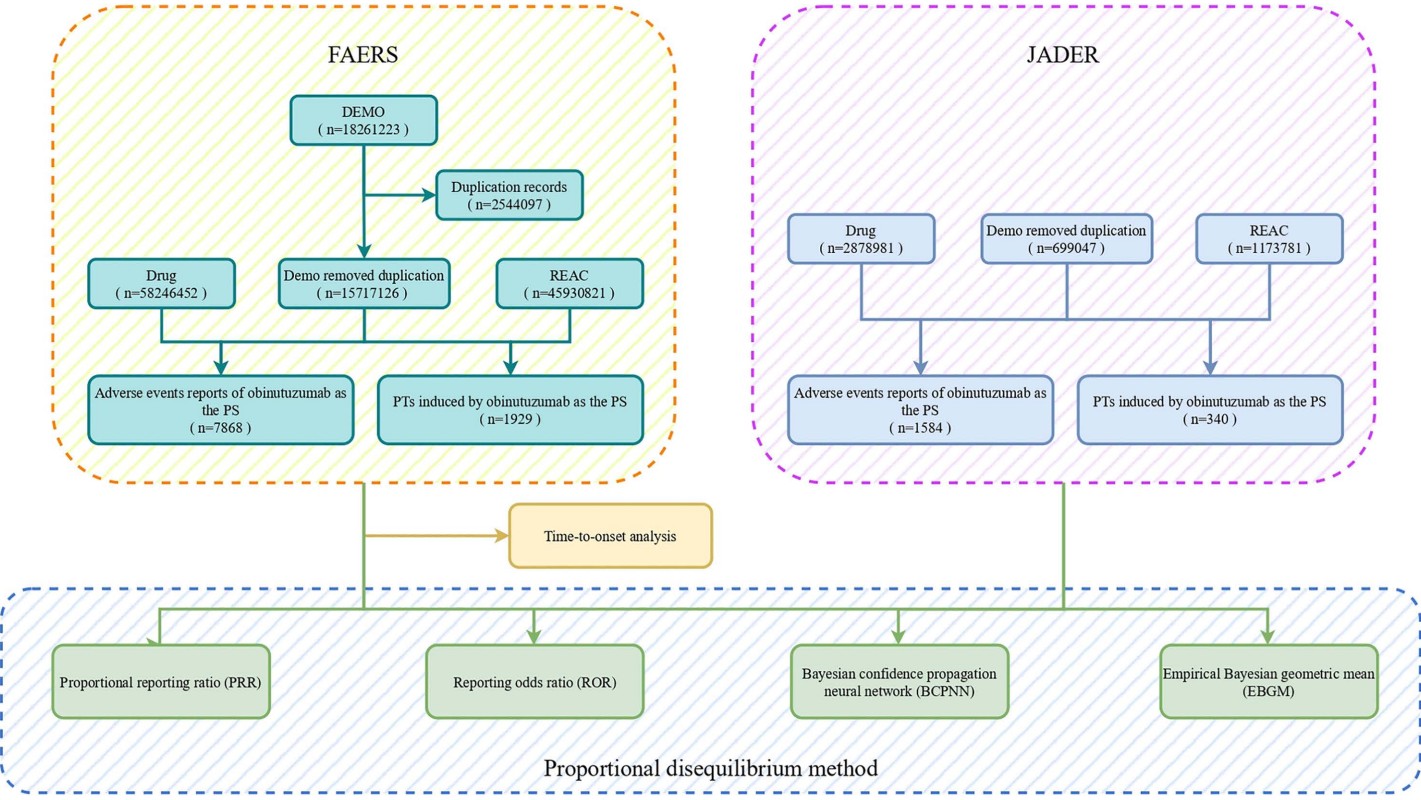

**Fig 1. The flowchat of extracting and analyzing obinutuzumab-related AEs from FAERS and JADER databases.** AEs, adverse events; FAERS, Food and Drug Administration Adverse Event Reporting System; JADER, Japanese Adverse Drug Event Report.

reports was 2019, with a total of 293 cases. Among all reported cases, males accounted for 49.85% (n = 4,712), and females accounted for 36.85% (n = 3,483), resulting in a male-to-female ratio of approximately 1.35:1. FL was the main indication (31.69%, n = 3,010), followed by chronic lymphocytic leukaemia (29.10%, n = 2,764). The reports in the FAERS database originated from multiple countries, while all data in the JADER database were from Japan. Overall, Japan had the highest number of reported cases (34.51%, n = 2,724), followed by the United States (29.17%, n = 2,303). Elderly patients (≥65 years old) constituted the highest proportion. Table 2 presents the specific numbers and proportions of these cases, while Fig 2 indicates that the majority of cases were reported in 2022 (16.6%, n = 1,268).

### 3.2. Detection of obinutuzumab signals

**3.2.1. Signals detection according to system organ class levels.** A total of 7,868 AE reports were retrieved from the FAERS database. These signals were then mapped to 16 SOCs. Among them, Blood and lymphatic system disorders accounted for the largest proportion (20.02%, n = 2,098), followed by infections and infestations (18.12%, n = 1,899), and general disorders and administration site conditions (16.76%, n = 1,756). In the JADER database, 1,584 AE reports were identified and subsequently mapped to 8 SOCs. Infections and infestations constituted the largest proportion (40.20%, n = 679), followed by blood and lymphatic system disorders (32.27%, n = 545), and investigations (22.97%, n = 388). The distribution of other SOCs is presented in Fig 3.

**3.2.2. Signals detection based on preferred terms level.** The effective obinutuzumab signals retrieved from the FAERS and JADER databases were sorted in descending order according to the ROR signal value and the number of occurrence cases. This was done to identify relevant and frequently observed AE signals, as presented in Tables 3 and 4. The study explored AE signals by excluding cases with less than 3 occurrences, drug indications, and disease progression. Among the high-intensity signals in FAERS databases, disseminated enteroviral infection (ROR = 1394.41) ranked first, followed by meningitis enteroviral (ROR = 129.11), post-acute COVID-19 syndrome (ROR = 80.35), and sinusitis aspergillus (ROR = 69.72). The most frequently reported signals included infusion-related reaction (ROR = 30.54), myelosuppression (ROR = 31.76), and tumour lysis syndrome (ROR = 62.41). In JADER, sinobronchitis emerged as the high-intensity signal with an ROR of 984.45. CD4 lymphocytes decreased showed a significant association with an ROR of 217.22, along with post-acute COVID-19 syndrome (ROR = 210.88). The common signal observed was myelosuppression, with a total count of 298 cases, followed by COVID-19 (n = 212) and COVID-19 pneumonia (n = 88). Among the top 30 PTs ranked, 11 identical AEs were found in both databases. In addition, the study identified several AEs not described in the manual, such as cytomegalovirus infection, myelosuppression, myelonecrosis, bronchiolitis obliterans, listeria septicemia, and hypogammaglobulinemia.

**3.2.3. Relationship between obinutuzumab related AEs and duration of administration.** The occurrence time of AEs induced by obinutuzumab in patients was determined based on the START_DT and EVENT_DT variables in the FAERS database. A total of 3,269 reports provided both the initiation time and occurrence time of AEs. Calculation revealed that the median time to AE onset following obinutuzumab treatment was 14 days, and 44.60% (n = 1,700) of patients experienced adverse reactions within 7 days of drug administration. As depicted in Fig 4, 57.98% (n = 2,210) of patients experienced AEs within 30 days of medication.

**3.2.4. Comparison of AEs caused by obinutuzumab in tumor indications and non-tumor indications.** Obinutuzumab has a rather extensive range of indications, encompassing those for hematological malignancies, lupus nephritis, membranous glomerulonephritis, lupus erythematosus, nephrotic syndrome, and so on. When obinutuzumab is employed to treat different diseases, the AEs induced by it may vary. Given that there is extremely limited data on the non-tumor indications of obinutuzumab in the JADER database, we solely analyzed the differences in AEs caused by obinutuzumab for tumor and non-tumor indications in the FAERS database (Figs 5 and 6). When obinutuzumab is used for the treatment of hematological malignancies, the most frequently reported AEs include COVID-19 infection (ROR = 8.44), followed by IRR (ROR = 29.42), pyrexia (ROR = 5.30), and

**Table 2. Clinical characteristics of reports of obinutuzumab from the FAERS and JADER databases.**

| Characteristics | FAERS | | JADER | | Total | |
|---|---|---|---|---|---|---|
| | N | Ratio (%) | N | Ratio (%) | N | Ratio (%) |
| **Number of events** | 7,868 | 83.24% | 1584 | 16.76% | 9,452 | 100.00% |
| **Gender** | | | | | | |
| Female | 2,731 | 34.71% | 752 | 47.47% | 3,483 | 36.85% |
| Male | 3,983 | 50.62% | 729 | 46.02% | 4,712 | 49.85% |
| Unknown | 1,154 | 14.67% | 103 | 6.50% | 1,257 | 13.30% |
| **Age** | | | | | | |
| <18 | 30 | 0.38% | | | 30 | 0.32% |
| 18≤and≤65 | 2,465 | 31.33% | 850 | 53.66% | 3,315 | 35.07% |
| >65 | 3,075 | 39.08% | 689 | 43.50% | 3,764 | 39.82% |
| Unknown | 2,298 | 29.21% | 45 | 2.84% | 2,343 | 24.79% |
| **Indications (TOP twenty)[a]** | | | | | | |
| Follicular lymphoma | 1,608 | 20.43% | 1402 | 84.04% | 3,010 | 31.69% |
| Chronic lymphocytic leukaemia | 2,764 | 35.12% | | | 2,764 | 29.10% |
| B-cell lymphoma | 641 | 8.14% | 16 | 3.12% | 690 | 7.31% |
| Diffuse large B-cell lymphoma | 419 | 5.32% | 9 | 0.55% | 428 | 4.51% |
| Non-Hodgkin's lymphoma | 186 | 2.36% | | | 186 | 1.96% |
| Lymphoma | 136 | 1.73% | | | 136 | 1.43% |
| Mantle cell lymphoma | 129 | 1.64% | | | 129 | 1.36% |
| Non-Hodgkin's lymphoma unspecified histology indolent | 96 | 1.22% | | | 96 | 1.01% |
| Glomerulonephritis membranous | 59 | 0.75% | | | 59 | 0.62% |
| Lupus nephritis | 58 | 0.74% | 5 | 0.31% | 63 | 0.66% |
| Nephrotic syndrome | 39 | 0.50% | | | 39 | 0.41% |
| Waldenstrom's macroglobulinaemia | 35 | 0.44% | | | 35 | 0.37% |
| Marginal zone lymphoma | 33 | 0.42% | | | 33 | 0.35% |
| B-cell lymphoma refractory | 31 | 0.39% | 3 | 0.18% | 34 | 0.36% |
| Chronic lymphocytic leukaemia refractory | 27 | 0.34% | 48 | 2.96 | 75 | 0.79% |
| Follicular lymphoma stage IV | 26 | 0.33% | | | 26 | 0.27% |
| Systemic lupus erythematosus | 25 | 0.32% | | | 25 | 0.26% |
| Lymphocytic lymphoma | 25 | 0.32% | | | 25 | 0.26% |
| Chronic lymphocytic leukaemia recurrent | 24 | 0.30% | | | 24 | 0.25% |
| Recurrent B-cell lymphoma | 19 | 0.24% | | | 19 | 0.20% |
| Other | 463 | 5.88% | 5 | 0.31% | 468 | 4.93% |
| Unknown | 1047 | 13.30% | 138 | 8.52% | 159 | 12.48% |
| **Reported Countries** | | | | | | |
| America | 2303 | 29.27% | | | 2303 | 24.37% |
| Japan | 1140 | 14.49% | 1584 | 100.00% | 2724 | 28.82% |
| France | 496 | 6.30% | | | 496 | 5.25% |
| China | 757 | 9.62% | | | 757 | 8.01% |
| Germany | 449 | 5.71% | | | 449 | 4.75% |
| Other | 2723 | 34.61% | | | 2723 | 28.81% |
| **Outcome[b]** | | | | | | |
| Hospitalization | 2929 | 32.17% | | | 2929 | 25.27% |
| Death | 955 | 10.49% | 205 | 8.25% | 1160 | 10.01% |
| Life-threatening | 358 | 3.93% | | | 358 | 3.09% |
| Disability | 79 | 0.87% | | | 79 | 0.68% |

*(Continued)*

**Table 2.** (Continued)

| Characteristics | FAERS | | JADER | | Total | |
|---|---|---|---|---|---|---|
| | N | Ratio (%) | N | Ratio (%) | N | Ratio (%) |
| Required intervention to prevent permanent impairment/damage | 20 | 0.22% | | | 20 | 0.17% |
| Congenital Anomaly | 1 | 0.01% | | | 1 | 0.01% |
| Other serious medical events | 2894 | 31.78% | | | 2894 | 24.97% |
| Not healed | | | 181 | 7.29% | 181 | 1.56% |
| Healed | | | 1296 | 52.17% | 1296 | 11.18% |
| Unknown | 1869 | 20.53% | 802 | 32.29% | 2671 | 23.05% |
| **Reported Person[c]** | | | | | | |
| Health profession | 7321 | 93.05% | 1693 | 106.88% | 9014 | 94.20% |
| Non-healthcare profession | 518 | 6.58% | 8 | 0.51% | 526 | 5.50% |
| Unknown | 29 | 0.37% | | | 29 | 0.30% |

FAERS, Food and Drug Administration Adverse Event Reporting System; JADER, Japanese Adverse Drug Event Report. a and b: Multiple outcomes may be involved in the same AE report, so this item is counted as "examples". c: The same AE report in the JADER database may involve multiple identities. N, the number of reports.

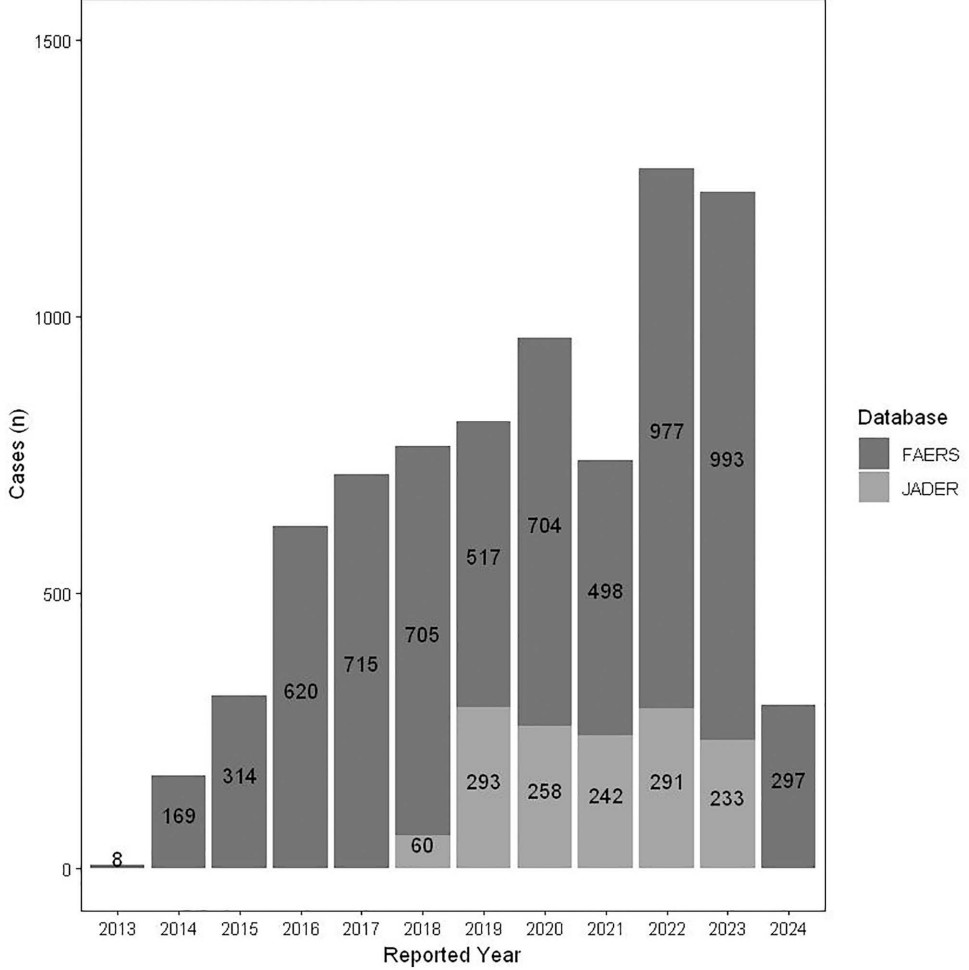

**Fig 2. Annual distribution of AE reports of obinutuzumab.** AE, adverse event; FAERS, Food and Drug Administration Adverse Event Reporting System; JADER, Japanese Adverse Drug Event Report.

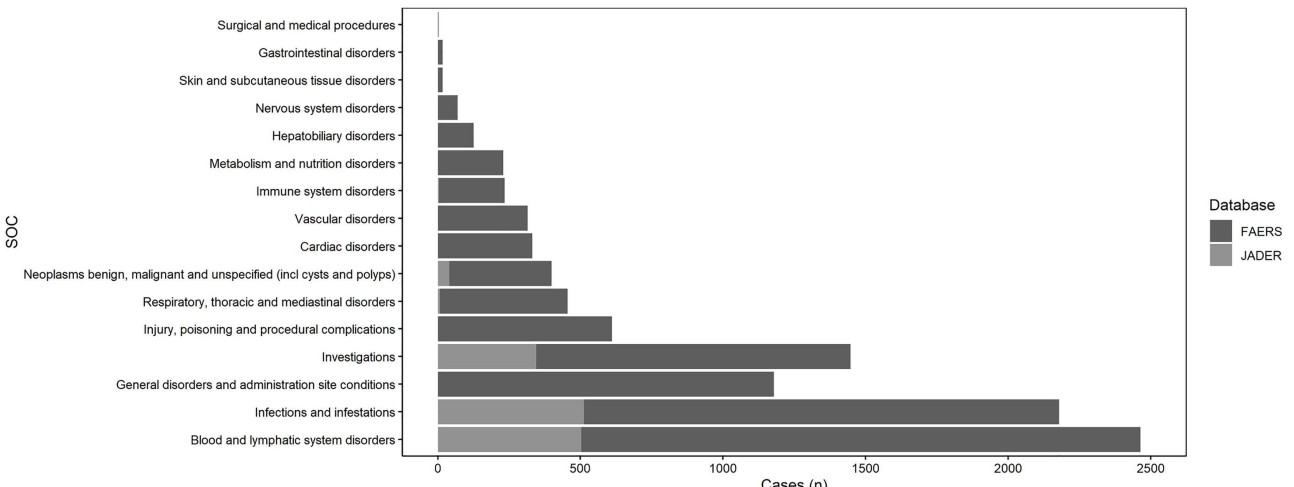

**Fig 3. The reported cases of AEs at each SOC level.** AEs, adverse events; SOC, system organ class; FAERS, Food and Drug Administration Adverse Event Reporting System; JADER, Japanese Adverse Drug Event Report.

neutropenia (ROR = 10.03). In the ranking of signal strengths within the FAERS database, disseminated enteroviral infection (ROR = 1660.83) ranks first, followed by meningitis enteroviral (ROR = 153.78), post-acute COVID-19 syndrome (ROR = 85.07), and sinusitis aspergillus (ROR = 83.04). Regarding non-tumor indications, the most frequently reported AEs are neutropenia (ROR = 11.24), chills (ROR = 13.74), pyrexia (ROR = 3.63), and hypotension (ROR = 5.36). In the ROR ranking, post-acute COVID-19 syndrome (ROR = 55.63) takes precedence, followed by aspergillus infection (ROR = 52.07), cytopenia (ROR = 45.26), and hypogammaglobulinaemia (ROR = 31.97). Among the top 20 risk signals ranked by occurrence frequency, 14 are identical. However, in the ROR ranking, only 3 AEs are the same. As shown in Fig 5, hypogammaglobulinemia (ROR = 31.97), cytomegalovirus chorioretinitis (ROR = 31.78), and cytomegalovirus viraemia (ROR = 17.93) are not specified in the drug instruction. These AEs also appear in the statistics of tumor indications. When obinutuzumab is employed for tumor indications, the incidence of AEs associated with infections and IRRs is higher, and the signals are more pronounced. Overall, regardless of whether obinutuzumab is used for tumor or non-tumor indications, AEs mainly consist of infections, blood and lymphatic system disorders, and IRRs.

## 4. Discussion

Obinutuzumab represents a new generation of type II CD20 monoclonal antibodies. The Fc segment of obinutuzumab undergoes glycoylation modification, which enhances its affinity with immune effector cells and demonstrates superior anti-tumor activity in both in vivo and in vitro studies [19,20]. Multiple clinical studies have indicated that the obinutuzumab regimen extends PFS and improves the quality of life in patients with newly diagnosed FL. Moreover, the AEs associated with this treatment are generally well-tolerated. Notwithstanding, it remains crucial to closely monitor the real-world use and AEs of obinutuzumab to guarantee its safety and efficacy. This study systematically evaluated the AEs associated with obinutuzumab by comprehensively analyzing data from the FAERS and the JADER databases from July 2013–2025. Through this process, the study verified existing safety information and identified novel potential risks. This provides a more comprehensive assessment of the potential risks of the drug in clinical use and serves as a safety reference for clinical practice. The following is a detailed discussion of the study findings.

**Table 3. The top 30 PTs ranked by ROR in FAERS database.**

| SOC | PT | N | ROR (95% CI) | PRR (χ2) | IC (IC025) | EBGM (EBGM05) |
|---|---|---|---|---|---|---|
| **Infections and infestations** | Disseminated enteroviral infection | 3 | 1394.41 (333.21- 5835.28) | 1394.20(2610.38) | 9.77(2.33) | 871.75(208.32) |
| | Meningitis enteroviral | 3 | 129.11 (40.37-412.97) | 129.09(361.23) | 6.93(2.17) | 122.35(38.25) |
| | Post-acute COVID-19 syndrome | 36 | 80.35 (57.62-112.05) | 80.20(2721.88) | 6.28(4.50) | 77.56(55.62) |
| | Sinusitis aspergillus | 3 | 69.72 (22.11-219.87) | 69.71(197.26) | 6.08(1.93) | 67.71(21.47) |
| | Cytomegalovirus enterocolitis | 15 | 65.57 (39.24-109.55) | 65.52(926.86) | 5.99(3.59) | 63.75(38.15) |
| | Coronavirus pneumonia | 3 | 60.10(19.10-189.11) | 60.09(169.94) | 5.87(1.87) | 58.61(18.63) |
| | Cytomegalovirus chorioretinitis | 35 | 59.51(42.54-83.26) | 59.41(1959.80) | 5.86(4.19) | 57.95(41.42) |
| | Cytomegalovirus gastroenteritis | 3 | 53.22(16.94-167.18) | 53.21(150.26) | 5.70(1.82) | 52.04(16.57) |
| | Cytomegalovirus hepatitis | 4 | 36.31(13.52-97.51) | 36.31(135.23) | 5.16(1.92) | 35.76(13.32) |
| | COVID-19 pneumonia | 158 | 34.35(29.34-40.22) | 34.08(5001.56) | 5.07(4.33) | 33.60(28.70) |
| | Pneumonia haemophilus | 4 | 26.26(9.80-70.37) | 26.26(96.09) | 4.70(1.75) | 25.97(9.69) |
| | Superinfection bacterial | 6 | 22.60(10.11-50.51) | 22.60(122.65) | 4.48(2.01) | 22.39(10.02) |
| | Campylobacter gastroenteritis | 4 | 19.74(7.38-52.81) | 19.73(70.54) | 4.29(1.60) | 19.58(7.32) |
| | Campylobacter infection | 5 | 19.21(7.97-46.32) | 19.20(85.57) | 4.25(1.76) | 19.05(7.90) |
| | Neutropenic infection | 5 | 18.02(7.47-43.44) | 18.01(79.73) | 4.16(1.73) | 17.88(7.42) |
| **Blood and lymphatic system disorders** | Bone marrow necrosis | 3 | 51.64(16.45-162.16) | 51.64(145.73) | 5.66(1.80) | 50.54(16.09) |
| | Myelosuppression | 317 | 31.76(28.41-35.52) | 31.27(9169.83) | 4.95(4.43) | 30.87(27.60) |
| | Cytopenia | 101 | 24.33(19.99-29.62) | 24.21(2224.92) | 4.58(3.77) | 23.97(19.70) |
| | Aplasia pure red cell | 15 | 17.85(10.74-29.67) | 17.84(236.60) | 4.15(2.49) | 17.71(10.65) |
| **Investigations** | Cytomegalovirus test positive | 16 | 31.29(19.10-51.26) | 31.27(462.60) | 4.95(3.02) | 30.87(18.84) |
| | Human rhinovirus test positive | 3 | 23.40(7.50-72.96) | 23.39(63.67) | 4.53(1.45) | 23.17(7.43) |
| | Procalcitonin increased | 9 | 22.79(11.82-43.95) | 22.78(185.61) | 4.50(2.33) | 22.57(11.70) |
| | CD4 lymphocytes decreased | 12 | 20.98(11.88-37.04) | 20.97(226.12) | 4.38(2.48) | 20.79(11.77) |
| **Metabolism and nutrition disorders** | Tumour lysis syndrome | 167 | 62.41(53.49-72.82) | 61.89(9746.20) | 5.91(5.07) | 60.31(51.69) |
| | Hyperphosphataemia | 11 | 17.85(9.86-32.30) | 17.84(173.50) | 4.15(2.29) | 17.71(9.78) |
| **Immune system disorders** | Hypogammaglobulinae-mia | 47 | 24.08(18.06-32.11) | 24.03(1026.81) | 4.57(3.43) | 23.79(17.84) |
| | Humoral immune defect | 3 | 22.71(7.28-70.81) | 22.71(61.65) | 4.49(1.44) | 22.50(7.22) |
| **Injury, poisoning and procedural complications** | Infusion related reaction | 594 | 30.54(28.13-33.16) | 29.65(16256.11) | 4.87(4.49) | 29.29(26.98) |
| **Neoplasms benign, malignant and unspecified (incl cysts and polyps)** | Tumour flare | 6 | 30.52(13.64-68.29) | 30.51(169.03) | 4.91(2.20) | 30.13(13.46) |

AEs, adverse events; SOC, system organ class; PT, preferred term; ROR, reporting odds ratio; PRR, proportional reporting ratio; CI, confidence interval; χ², chi-squared; IC, information component; IC025, the lower bound of 95% CI; EBGM, empirical Bayesian geometric mean; EBGM05, the lower bound of 95% CI; N, the number of reports. FAERS, Food and Drug Administration Adverse Event Reporting System.

**Table 4. The top 30 PTs ranked by ROR in JADER database.**

| SOC | PT | N | ROR (95% CI) | PRR (χ2) | IC (IC025) | EBGM (EBGM05) |
|---|---|---|---|---|---|---|
| **Infections and infestations** | Sinobronchitis | 4 | 984.45(180.22-5377.47) | 982.78(1307.72) | 8.36(1.53) | 328.26(60.09) |
| | Post-acute COVID-19 syndrome | 20 | 210.88(124.77-356.43) | 209.10(2905.86) | 7.20(4.26) | 146.98(86.96) |
| | COVID-19 pneumonia | 88 | 82.83(65.88-104.15) | 79.78(5892.94) | 6.10(4.85) | 68.78(54.70) |
| | Cytomegalovirus chorioretinitis | 35 | 57.22(40.24-81.38) | 56.39(1708.72) | 5.66(3.98) | 50.69(35.64) |
| | Cytomegalovirus hepatitis | 6 | 50.96(21.97-118.22) | 50.83(265.65) | 5.53(2.38) | 46.16(19.90) |
| | Listeria sepsis | 4 | 41.89(15.08-116.36) | 41.82(146.88) | 5.27(1.90) | 38.62(13.90) |
| | COVID-19 | 212 | 38.12(32.95-44.09) | 34.78(6514.23) | 5.02(4.34) | 32.55(28.14) |
| | Pneumonia cytomegaloviral | 16 | 26.20(15.82-43.40) | 26.03(365.88) | 4.63(2.80) | 24.77(14.96) |
| | Progressive multifocal leukoencephalopathy | 19 | 23.17(14.60-36.78) | 23.00(382.02) | 4.46(2.81) | 22.01(13.87) |
| | Enterobacter bacteraemia | 6 | 18.59(8.22-42.04) | 18.54(95.97) | 4.16(1.84) | 17.91(7.92) |
| | Cytomegalovirus infection | 69 | 17.59(13.79-22.44) | 17.11(1013.00) | 4.05(3.17) | 16.57(12.99) |
| | Cytomegalovirus gastroenteritis | 21 | 16.31(10.54-25.24) | 16.17(289.60) | 3.97(2.57) | 15.69(10.14) |
| | Coronavirus infection | 10 | 16.18(8.60-30.41) | 16.11(137.27) | 3.97(2.11) | 15.63(8.32) |
| | Hepatitis B | 7 | 12.14(5.73-25.74) | 12.11(69.66) | 3.57(1.68) | 11.84(5.59) |
| | Cytomegalovirus viraemia | 14 | 10.56(6.21-17.96) | 10.50(117.92) | 3.37(1.98) | 10.30(6.06) |
| | Hepatitis B reactivation | 18 | 10.06(6.30-16.08) | 9.99(142.91) | 3.30(2.06) | 9.82(6.14) |
| | Sinusitis | 3 | 9.28(2.96-29.11) | 9.27(21.73) | 3.19(1.02) | 9.12(2.91) |
| | Cytomegalovirus infection reactivation | 6 | 6.67(2.98-14.94) | 6.66(28.46) | 2.72(1.21) | 6.58(2.94) |
| **Blood and lymphatic system disorders** | Myelosuppression | 298 | 27.86(24.61-31.54) | 24.47(6425.05) | 4.55(4.02) | 23.36(20.63) |
| | Cytopenia | 29 | 11.59(8.00-16.78) | 11.46(270.73) | 3.49(2.41) | 11.22(7.74) |
| | Lymphopenia | 51 | 10.17(7.69-13.46) | 9.97(404.42) | 3.29(2.49) | 9.79(7.40) |
| | Haematotoxicity | 5 | 8.64(3.56-20.93) | 8.62(33.11) | 3.09(1.27) | 8.49(3.50) |
| **Investigations** | CD4 lymphocytes decreased | 11 | 217.22(106.76-441.96) | 216.21(1636.43) | 7.23(3.56) | 150.45(73.95) |
| | Cytomegalovirus test positive | 3 | 6.42(2.05-20.06) | 6.41(13.52) | 2.66(0.85) | 6.34(2.03) |
| **Respiratory, thoracic and mediastinal disorders** | Obliterative bronchiolitis | 8 | 41.09(19.95-84.61) | 40.95(287.83) | 5.24(2.55) | 37.88(18.39) |
| **Neoplasms benign, malignant and unspecified (incl cysts and polyps)** | Tumour associated fever | 3 | 33.55(10.41-108.11) | 33.50(88.57) | 4.97(1.54) | 31.43(9.75) |
| **Endocrine disorders** | Primary adrenal insufficiency | 4 | 22.37(8.21-60.99) | 22.34(77.98) | 4.42(1.62) | 21.41(7.85) |
| **Skin and subcutaneous tissue disorders** | Dermatitis acneiform | 7 | 11.85(5.59-25.11) | 11.82(67.72) | 3.53(1.67) | 11.57(5.46) |
| **Immune system disorders** | Hypogammaglobulinaemia | 5 | 7.24(2.99-17.52) | 7.23(26.44) | 2.84(1.17) | 7.14(2.95) |
| **Blood and lymphatic system disorders** | Aplastic anaemia | 6 | 6.07(2.71-13.58) | 6.05(25.02) | 2.58(1.15) | 5.99(2.68) |

AEs, adverse events; SOC, system organ class; PT, preferred term; ROR, reporting odds ratio; PRR, proportional reporting ratio; CI, confidence interval; χ², chi-squared; IC, information component; IC025, the lower bound of 95% CI; EBGM, empirical Bayesian geometric mean; EBGM05, the lower bound of 95% CI; N, the number of reports; JADER, Japanese Adverse Drug Event Report.

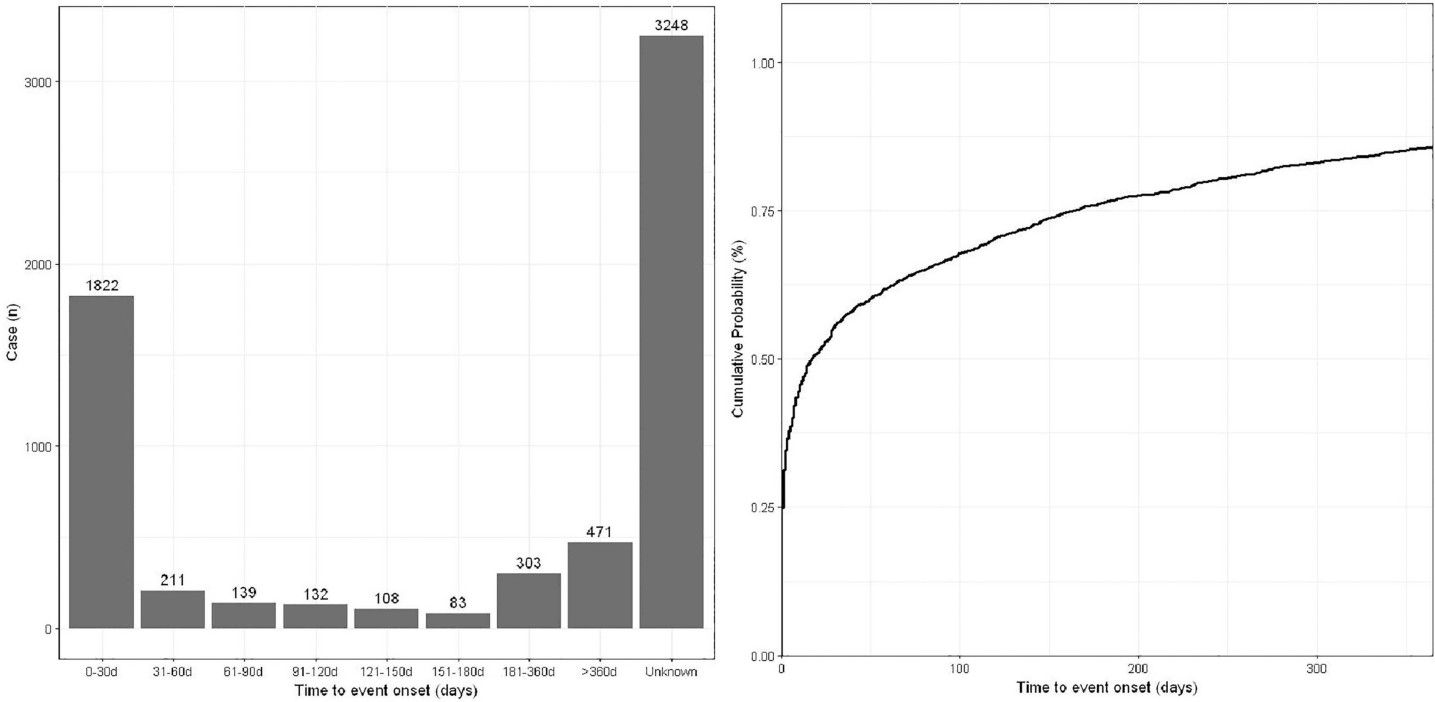

**Fig 4. Time to event onset of obinutuzumab-related AEs in FAERS database.** AEs, adverse events; SOC, system organ class; FAERS, Food and Drug Administration Adverse Event Reporting System.

### 4.1. Analysis of basic information on obinutuzumab related AEs

Among the 9,452 patients treated with obinutuzumab reported in this study, excluding data of unknown origin, the male-to-female ratio was 1.35:1, and patients aged over 65 years accounted for 39.82%. The clinical characteristics of AEs associated with obinutuzumab may be related to the incidence of lymphoma. It has been found that in developed countries, the incidence of NHL is approximately 1.29 times higher in males than in females [21]. Moreover, 57% of NHL patients are diagnosed at an age greater than 65 years [22]. Given the data sources in the two databases and the marketing status of obinutuzumab, the countries contributing reports were mainly the United States and Japan. The approved indications of obinutuzumab were generally consistent across countries, with no obvious off-label use. Obinutuzumab is formulated as an injection and is typically administered in hospitals. Consequently, the majority of the reporting population consists of medical staff, and the number of AEs voluntarily reported by non-healthcare individuals is relatively low.

### 4.2. Analysis of infection risk signals associated with obinutuzumab

Among the top 30 signal strengths of obinutuzumab-associated AEs, over half were infections and infestations. In the FAERS database, there were 15 AEs belonging to the category of infections and infestations, while in the JADER database, there were 18 such AEs. These AEs included sinusitis, COVID-19 infection, neutropenic infection, infectious diarrhea, viral hepatitis, bacterial co-infection, hepatitis B, coronavirus infection, and progressive multifocal leukoencephalopathy (PML), as documented in the drug instruction. Additionally, the infections and infestations category also encompassed several AEs not described in the drug instruction, such as cytomegalovirus (CMV) infection, campylobacter gastroenteritis, campylobacter infection, haemophilus pneumonia, enterobacter bacteremia, and hypogammaglobulinemia. When a large number of B cells are depleted, obinutuzumab may impact the immune function of T cells, thus

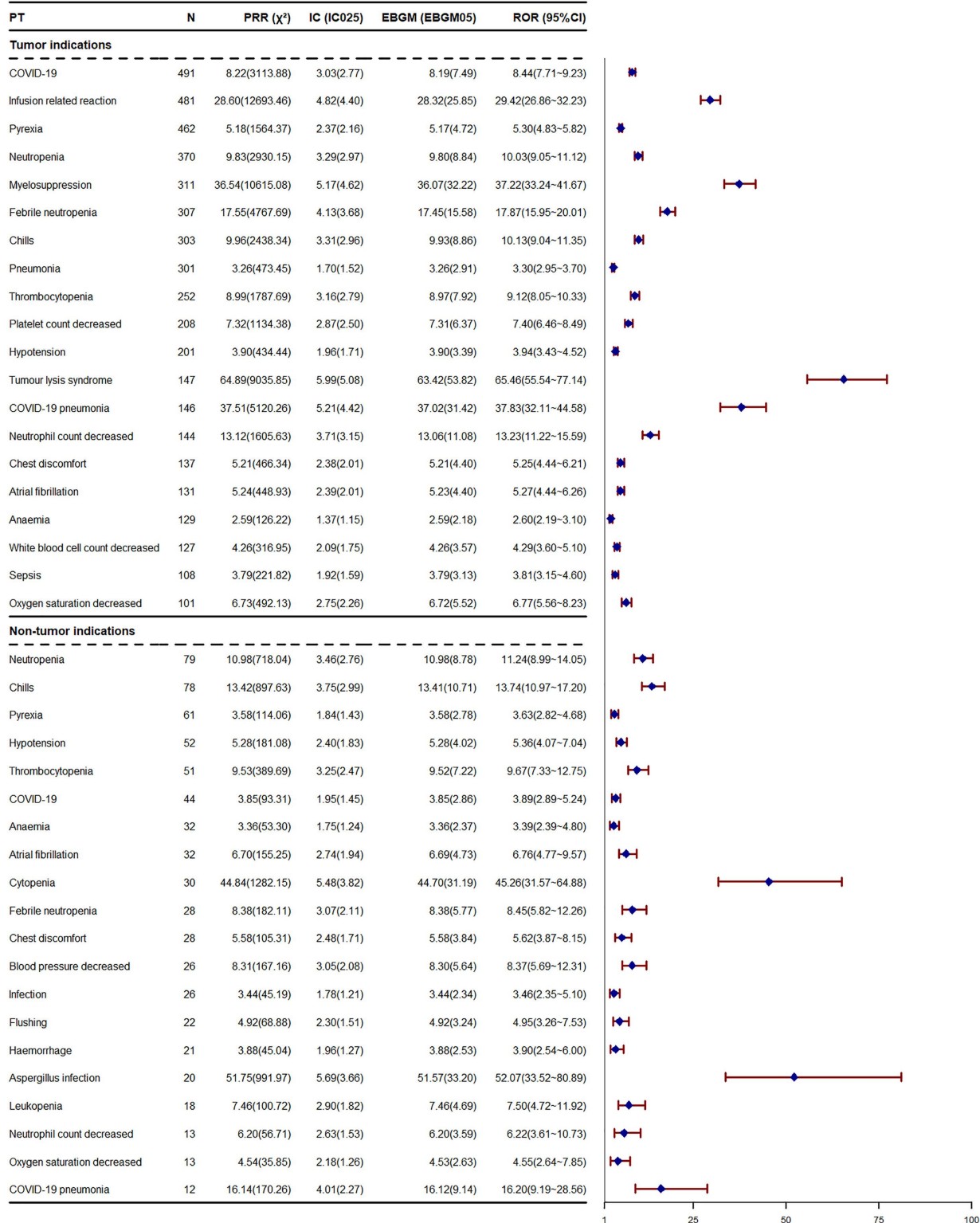

| PT | N | PRR ($\chi^2$) | IC (IC025) | EBGM (EBGM05) | ROR (95%CI) |
|---|---|---|---|---|---|
| **Tumor indications** | | | | | |
| COVID-19 | 491 | 8.22(3113.88) | 3.03(2.77) | 8.19(7.49) | 8.44(7.71~9.23) |
| Infusion related reaction | 481 | 28.60(12693.46) | 4.82(4.40) | 28.32(25.85) | 29.42(26.86~32.23) |
| Pyrexia | 462 | 5.18(1564.37) | 2.37(2.16) | 5.17(4.72) | 5.30(4.83~5.82) |
| Neutropenia | 370 | 9.83(2930.15) | 3.29(2.97) | 9.80(8.84) | 10.03(9.05~11.12) |
| Myelosuppression | 311 | 36.54(10615.08) | 5.17(4.62) | 36.07(32.22) | 37.22(33.24~41.67) |
| Febrile neutropenia | 307 | 17.55(4767.69) | 4.13(3.68) | 17.45(15.58) | 17.87(15.95~20.01) |
| Chills | 303 | 9.96(2438.34) | 3.31(2.96) | 9.93(8.86) | 10.13(9.04~11.35) |
| Pneumonia | 301 | 3.26(473.45) | 1.70(1.52) | 3.26(2.91) | 3.30(2.95~3.70) |
| Thrombocytopenia | 252 | 8.99(1787.69) | 3.16(2.79) | 8.97(7.92) | 9.12(8.05~10.33) |
| Platelet count decreased | 208 | 7.32(1134.38) | 2.87(2.50) | 7.31(6.37) | 7.40(6.46~8.49) |
| Hypotension | 201 | 3.90(434.44) | 1.96(1.71) | 3.90(3.39) | 3.94(3.43~4.52) |
| Tumour lysis syndrome | 147 | 64.89(9035.85) | 5.99(5.08) | 63.42(53.82) | 65.46(55.54~77.14) |
| COVID-19 pneumonia | 146 | 37.51(5120.26) | 5.21(4.42) | 37.02(31.42) | 37.83(32.11~44.58) |
| Neutrophil count decreased | 144 | 13.12(1605.63) | 3.71(3.15) | 13.06(11.08) | 13.23(11.22~15.59) |
| Chest discomfort | 137 | 5.21(466.34) | 2.38(2.01) | 5.21(4.40) | 5.25(4.44~6.21) |
| Atrial fibrillation | 131 | 5.24(448.93) | 2.39(2.01) | 5.23(4.40) | 5.27(4.44~6.26) |
| Anaemia | 129 | 2.59(126.22) | 1.37(1.15) | 2.59(2.18) | 2.60(2.19~3.10) |
| White blood cell count decreased | 127 | 4.26(316.95) | 2.09(1.75) | 4.26(3.57) | 4.29(3.60~5.10) |
| Sepsis | 108 | 3.79(221.82) | 1.92(1.59) | 3.79(3.13) | 3.81(3.15~4.60) |
| Oxygen saturation decreased | 101 | 6.73(492.13) | 2.75(2.26) | 6.72(5.52) | 6.77(5.56~8.23) |
| **Non-tumor indications** | | | | | |
| Neutropenia | 79 | 10.98(718.04) | 3.46(2.76) | 10.98(8.78) | 11.24(8.99~14.05) |
| Chills | 78 | 13.42(897.63) | 3.75(2.99) | 13.41(10.71) | 13.74(10.97~17.20) |
| Pyrexia | 61 | 3.58(114.06) | 1.84(1.43) | 3.58(2.78) | 3.63(2.82~4.68) |
| Hypotension | 52 | 5.28(181.08) | 2.40(1.83) | 5.28(4.02) | 5.36(4.07~7.04) |
| Thrombocytopenia | 51 | 9.53(389.69) | 3.25(2.47) | 9.52(7.22) | 9.67(7.33~12.75) |
| COVID-19 | 44 | 3.85(93.31) | 1.95(1.45) | 3.85(2.86) | 3.89(2.89~5.24) |
| Anaemia | 32 | 3.36(53.30) | 1.75(1.24) | 3.36(2.37) | 3.39(2.39~4.80) |
| Atrial fibrillation | 32 | 6.70(155.25) | 2.74(1.94) | 6.69(4.73) | 6.76(4.77~9.57) |
| Cytopenia | 30 | 44.84(1282.15) | 5.48(3.82) | 44.70(31.19) | 45.26(31.57~64.88) |
| Febrile neutropenia | 28 | 8.38(182.11) | 3.07(2.11) | 8.38(5.77) | 8.45(5.82~12.26) |
| Chest discomfort | 28 | 5.58(105.31) | 2.48(1.71) | 5.58(3.84) | 5.62(3.87~8.15) |
| Blood pressure decreased | 26 | 8.31(167.16) | 3.05(2.08) | 8.30(5.64) | 8.37(5.69~12.31) |
| Infection | 26 | 3.44(45.19) | 1.78(1.21) | 3.44(2.34) | 3.46(2.35~5.10) |
| Flushing | 22 | 4.92(68.88) | 2.30(1.51) | 4.92(3.24) | 4.95(3.26~7.53) |
| Haemorrhage | 21 | 3.88(45.04) | 1.96(1.27) | 3.88(2.53) | 3.90(2.54~6.00) |
| Aspergillus infection | 20 | 51.75(991.97) | 5.69(3.66) | 51.57(33.20) | 52.07(33.52~80.89) |
| Leukopenia | 18 | 7.46(100.72) | 2.90(1.82) | 7.46(4.69) | 7.50(4.72~11.92) |
| Neutrophil count decreased | 13 | 6.20(56.71) | 2.63(1.53) | 6.20(3.59) | 6.22(3.61~10.73) |
| Oxygen saturation decreased | 13 | 4.54(35.85) | 2.18(1.26) | 4.53(2.63) | 4.55(2.64~7.85) |
| COVID-19 pneumonia | 12 | 16.14(170.26) | 4.01(2.27) | 16.12(9.14) | 16.20(9.19~28.56) |

**Fig 5. The top 20 PTs, used in both tumor and non-tumor indications, are ranked according to frequency in FAERS database.** PT, preferred term; ROR, reporting odds ratio; PRR, proportional reporting ratio; CI, confidence interval; $\chi^2$, chi-squared; IC, information component; IC025, the lower bound of 95% CI; EBGM, empirical Bayesian geometric mean; EBGM05, the lower bound of 95% CI; N, the number of reports; FAERS, Food and Drug Administration Adverse Event Reporting System.

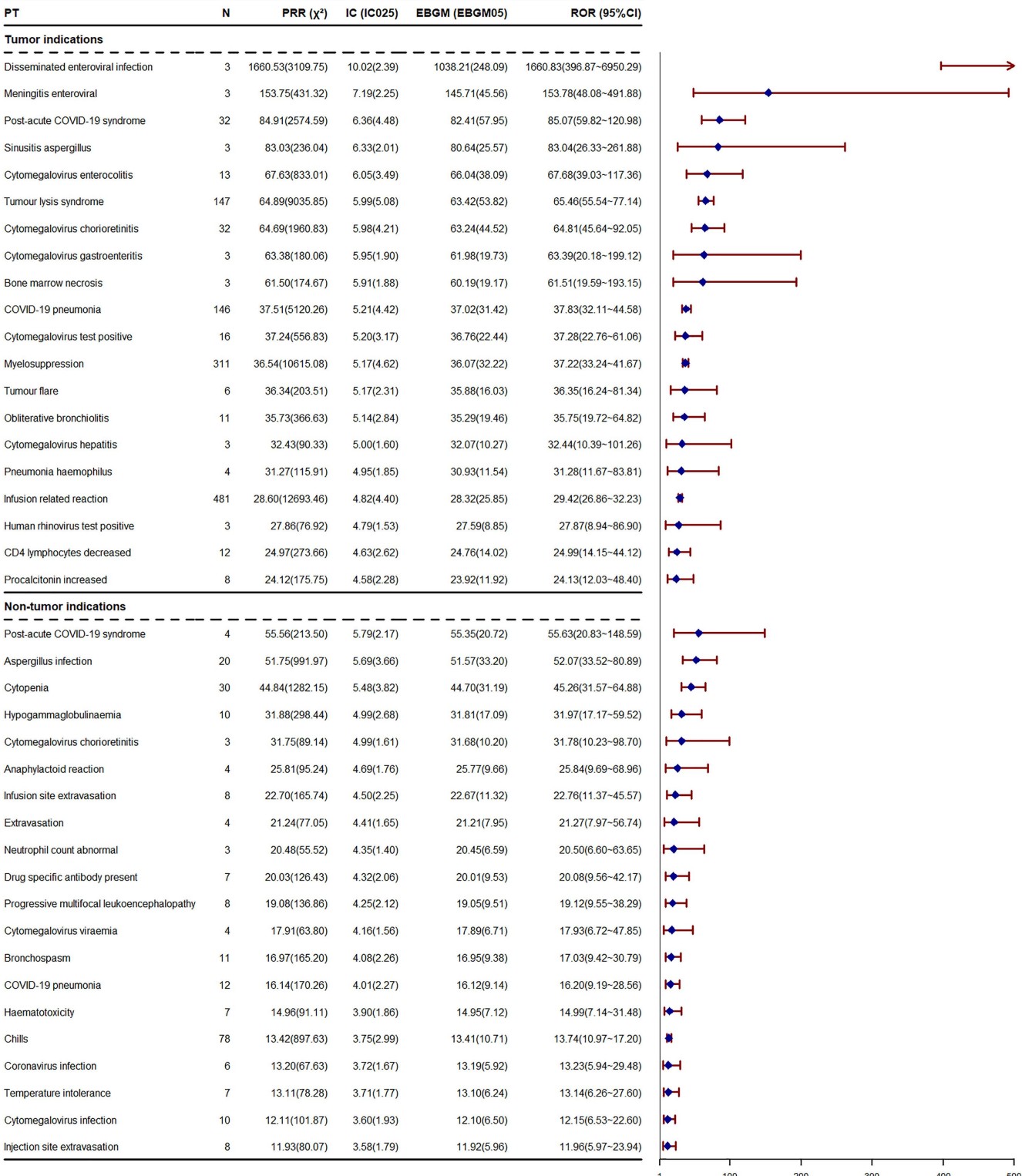

**Fig 6. The top 20 PTs, used in both tumor and non-tumor indications, are ranked by ROR in FAERS database.** PT, preferred term; ROR, reporting odds ratio; PRR, proportional reporting ratio; CI, confidence interval; $\chi^2$, chi-squared; IC, information component; IC025, the lower bound of 95% CI; EBGM, empirical Bayesian geometric mean; EBGM05, the lower bound of 95% CI; N, the number of reports; FAERS, Food and Drug Administration Adverse Event Reporting System.

increasing the risk of infection in patients. Patients treated with obinutuzumab have compromised immune function due to both the disease itself and drug-related factors, and are susceptible to bacterial, fungal, or viral infections. Among these, the most common clinical manifestations are granulocytosis with fever, lung infection, and urinary tract infection [23]. Compared with rituximab, obinutuzumab exhibits a more potent effect in eliminating B cells. Consequently, the use of obinutuzumab may elevate the risk of infection [3,24]. Amitai et al. [7] performed a meta-analysis of five randomized controlled trials (RCTs) comparing rituximab and obinutuzumab. The findings indicated that the treatment regimen incorporating obinutuzumab led to a significant increase in the incidence of grade 3–4 infections. Notably, the rates of grade 3–4 infections varied substantially across different studies, ranging from 3% to 20%. This variation is likely attributed to differences in the study populations. Irrespective of whether rituximab or obinutuzumab is employed, the infection rate appears to be predominantly influenced by other factors, such as the disease type, comorbidities, and the chemotherapy regimen utilized. Furthermore, in a retrospective cohort study, patients treated with obinutuzumab had a greater risk of prolonged SARS-CoV-2 infection and severe COVID-19 compared to those treated with rituximab [25].

When the US FDA approved obinutuzumab in 2013, it issued two black box warnings. One was for hepatitis B virus (HBV) reactivation, and the other was for PML. The typical clinical manifestation of HBV reactivation is an elevation in aminotransferase levels. In severe cases, this can result in fulminant hepatitis, liver failure, and even death [26]. In the JADER database, the risk signal of hepatitis B reactivation ranked 23rd, but it did not make it into the top 30 in the FAERS database. Nevertheless, it cannot be overlooked because it may lead to serious or life-threatening consequences during drug treatment. PML (FAERS, ROR = 16.09; JADER, ROR = 23.17) is another AE that cannot be ignored. PML is an incurable and potentially fatal brain disease caused by the reactivation of John Cunningham virus (JCV) [27]. Although few cases of PML caused by obinutuzumab have been reported, the US FDA has issued a black box warning for this drug due to the high mortality rate associated with this adverse reaction and the lack of conventional treatment options [28]. The initial clinical manifestations of PML typically include neurological symptoms such as cognitive impairment, aphasia, motor or sensory deficits, and seizures. The pathogenesis of PML is mainly associated with the cellular immune response, hypogammaglobulinemia, and the absence of interferon gamma in the central nervous system [29]. Therefore, for patients with PML, the use of obinutuzumab should be promptly discontinued to prevent sustained immunosuppressive effects.

Among the top 30 infection risk signals ranked in this research, the infection caused by CMV accounted for the largest proportion and was not recorded in the drug label. CMV disease is triggered by the initial infection, reinfection, or reactivation of CMV, which belongs to the beta-herpesvirus subfamily [30]. In cases of immune suppression, such as hematopoietic stem cell transplantation, organ transplantation, acquired immunodeficiency syndrome (AIDS), steroid therapy, and chemotherapy, latent infections from childhood and latency within the host can lead to the reactivation of CMV and the development of the disease [30]. Additionally, CMV infection can result in pneumonia, enteritis, hepatitis, retinitis, and encephalitis [31]. In this study, CMV was found to cause various infections, including CMV gastroenteritis, CMV chorioretinitis, CMV enterocolitis, CMV hepatitis, CMV pneumonia, etc. These infections were detected in the FAERS and JADER databases with strong risk signals. A comparable study indicated that the duration of CMV infection was significantly prolonged after treatment with bendamustine combined with obinutuzumab, reaching 61.0 days (33.0–102.5) [32]. Thus, caution is necessary, especially for elderly patients at high risk of immunosuppression. Regular CD4 positive cell counts should be measured, and infection prevention measures should be implemented.

### 4.3. Analysis of blood and lymphatic system disorders risk signals associated with obinutuzumab

Among the top 30 AEs in this study, those related to the blood and lymphatic system accounted for the second-highest proportion. The AEs in the FAERS database include bone marrow necrosis (ROR = 51.64), myelosuppression (ROR = 31.76), cytopenia (ROR = 24.33), aplasia pure red cell (ROR = 17.85). In the JADER database, the AEs include myelosuppression (ROR = 27.86), cytopenia (ROR = 11.59), lymphopenia (ROR = 10.17), haematotoxicity (ROR = 8.64), and dysplastic anemia (ROR = 6.07). Among these, bone marrow necrosis and myelosuppression are risk signals not

documented in the drug label. Moreover, myelosuppression shows a strong risk signal in both databases. The main clinical manifestations of myelosuppression include neutropenia, thrombocytopenia, leukopenia, and decreased hemoglobin, making it a serious AE. The bone marrow suppression induced by obinutuzumab may be associated with its own immunomodulatory mechanism. The RORs of thrombocytopenia in the FAERS and JADER databases are 9.21 and 2.82, respectively. Thrombocytopenia must not be overlooked, as it can give rise to blood clotting disorders. The analysis by Amitai et al. [7] indicated that the overall risk of severe thrombocytopenia (grade ≥ 3) induced by obinutuzumab is at least twice that seen with rituximab. The precise mechanism through which obinutuzumab causes thrombocytopenia remains not entirely clear. The pathological mechanisms of thrombocytopenia caused by these two monoclonal antibodies might both be ascribed to cytokine release. In a small percentage of CLL patients, upon the first administration of obinutuzumab, pro-inflammatory cytokines, such as interleukin-6 (IL-6) and interleukin-8 (IL-8), are directly released [12]. In an earlier study, additional explanatory methods involve eliminating platelets expressing the CD20 antigen via immune mechanisms or activating the complement system through soluble CD20 antigens in the circulation, thus reducing the platelet count [33]. Another hypothesis posits that CD20 might be transferred to platelets, particularly when considering patients with B-cell tumors who have experienced thrombocytopenia associated with anti-CD20 antibodies [34]. The severe thrombocytopenia induced by obinutuzumab is most prevalent in the first cycle. The associated risk factors include high levels of circulating pathogens, high CD20 expression levels, splenomegaly, and bone marrow infiltration [12]. Compared with rituximab, obinutuzumab appears to trigger a more intense cytokine release syndrome, thereby increasing the incidence of IRR and thrombocytopenia [1,12,35]. In the GALLIUM trial, 11.4% of the patients treated with obinutuzumab developed thrombocytopenia, while the proportion was 7.5% for those treated with rituximab [6]. Among both groups, 6.1% (of patients treated with obinutuzumab) and 2.7% (of those treated with rituximab) experienced "severe" thrombocytopenia (grade ≥ 3). Consequently, during the administration of obinutuzumab, patients should undergo enhanced post-medication monitoring and regular laboratory tests.

### 4.4. Analysis of metabolism and nutrition disorders risk signals associated with obinutuzumab

In the FAERS database, risk signals related to metabolism and nutrition disorders included TLS (ROR = 62.41) and hyperphosphatemia (ROR = 17.85). However, no such risk signals were detected in the JADER database. In the drug label, TLS is characterized as the most severe adverse drug reaction and is more prevalent in patients with a high tumor burden and/or impaired kidney function. Similarly, the label for rituximab also refers to the adverse reaction of TLS. TLS is a metabolic disorder triggered by spontaneous or treatment-induced lysis of tumor cells, which can be clinically presented as hyperkalemia, hyperphosphatemia, hyperuricemia, and hypocalcemia [36]. Obinutuzumab, via its anti-CD20 effect, rapidly induces apoptosis and lysis of tumor cells. This process leads to the release of a substantial amount of cellular contents into the bloodstream, thereby triggering TLS [37]. TLS results in the release of a large quantity of intracellular nucleic acid. Under the action of xanthine oxidase, this nucleic acid is converted into xanthine, which then produces uric acid. Intracellular potassium ions and phosphates are also released in large amounts. This indirectly causes calcium ion deposition and ultimately results in electrolyte imbalance [38]. In the GREEN study, the overall incidence of TLS was 6.4% among patients with CLL who were treated with obinutuzumab either alone or in combination with chemotherapy [39]. TLS can cause renal artery vasoconstriction, perfusion injury, and the deposition of calcium phosphate and uric acid. Therefore, prior to the first infusion of obinutuzumab, appropriate measures such as adequate hydration and the administration of urate-inhibiting drugs (e.g., allopurinol) or urate-oxidase (e.g., rasburicase) should be taken, to reduce the risk of TLS.

### 4.5. Analysis of immune system disorders signals associated with obinutuzumab

In this study, AEs associated with immune system disorders were found to include hypogammaglobulinaemia (FAERS, ROR = 24.08; JADER, ROR = 7.24), cytokine release syndrome (FAERS, ROR = 15.53), secondary immunodeficiency

(FAERS, ROR = 12.25), serum sickness (FAERS, ROR = 9.84), anaphylactoid reaction (FAERS, ROR = 8.28), anaphylactic shock (FAERS, ROR = 6.89), immunodeficiency (FAERS, ROR = 3.95). Among these, hypogammaglobulinaemia and anaphylactic shock were not documented in the drug instruction. Hypersensitivity reactions, leading to rapid anaphylaxis and delayed onset, have been reported following obinutuzumab administration. Moreover, a new risk signal for anaphylactic shock has been identified in this study. Similarly, the label for rituximab also mentions hypersensitivity reactions and immediate allergic reactions. However, the mechanism by which these two monoclonal antibodies induce allergic reactions remains unclear. Although obinutuzumab may trigger allergic reactions in non-human primates through an immune-complex-mediated mechanism, this is usually not predictable in humans [40]. Clinically, it is challenging to differentiate anaphylaxis from infusion-related reactions. Therefore, patients should be equipped with emergency measures and vigilant about the risk of anaphylactic shock during medication.

### 4.6. Analysis of IRRs signals associated with obinutuzumab

IRRs induced by obinutuzumab may occur during or after the infusion, with a higher prevalence, particularly during the first infusion [1]. These reactions can present with a diverse range of symptoms, including but not limited to nausea, chills, fever, hypotension, dyspnea, and laryngeal edema [41]. In the FAERS database, IRRs were found to be the most common AE, and the ROR reached 30.54. Amitai et al. [7] reported that the overall risk of severe IRR associated with obinutuzumab is at least twice as high as that observed with rituximab. The IRRs induced by rituximab and obinutuzumab may be associated with cytokine release. It was found that patients who experienced IRRs released higher levels of interleukin IL-6, IL-8, and tumor necrosis factor alpha (TNF-α) compared to those who did not [12]. This typically occurs during the first infusion and is accompanied by the rapid destruction of circulating B cells and a reduction in the number of circulating natural killer (NK) cells. To minimize the incidence and severity of IRRs caused by obinutuzumab, a series of preventive measures are commonly implemented in clinical practice. For example, patients may be pre-treated with medications such as antihistamines, glucocorticoids, and analgesics prior to receiving the obinutuzumab infusion, aiming to decrease the incidence and severity of IRRs [42]. Thus, IRRs caused by obinutuzumab are serious AEs that demand careful attention. Clinicians should have a comprehensive understanding of their mechanism.

### 4.7. Analysis of the efficacy and safety of obinutuzumab in non-tumor indications

The non-tumor indications of obinutuzumab mainly encompass membranous glomerulonephritis, lupus nephritis, nephrotic syndrome, Waldenström macroglobulinemia, and systemic lupus erythematosus. At present, although obinutuzumab has not yet been approved for non-tumor indications, animal experiments and clinical trials have been conducted to evaluate its efficacy and safety for these indications. The results indicate that its efficacy is even superior to that of rituximab. For instance, in a murine model of lupus nephritis, it was discovered that the use of obinutuzumab yielded better results than rituximab [43]. Another study revealed that obinutuzumab is effective for patients with renal and non-renal systemic lupus erythematosus who have developed secondary resistance to rituximab, and it can also reduce the disease activity rate [44]. Furie et al. [45] reported the findings of a clinical trial involving 125 lupus nephritis patients who had undergone standard treatment. These patients were randomly allocated to receive either placebo or obinutuzumab on the 1st day, the 2nd week, the 24th week, and the 26th week. The results showed that, compared with placebo treatment, obinutuzumab led to a higher complete renal remission rate at the 52nd week, and the improvement in renal response persisted until the 104th week. Regarding safety, obinutuzumab did not elevate the incidence of severe infections, severe AEs, or mortality. Recently, the REGENCY study (a phase III clinical trial) revealed that the combination of obinutuzumab and standard treatment (mycophenolate mofetil and glucocorticoids) could significantly increase the complete renal response rate (CRR) in lupus nephritis patients. At 76 weeks, the CRR in the obinutuzumab group was 46.4%, while that in the placebo group was 33.1%, and the difference was statistically significant (P = 0.0232) [46]. No unexpected safety signals were detected in the clinical trial. The serious AEs mainly included infections and events related to COVID-19, which were more

prevalent in the obinutuzumab group than in the placebo group. The clinical trial identified six serious AEs that occurred in five or more patients, namely COVID-related pneumonia, pneumonia, urinary tract infection, COVID virus infection, gastroenteritis, and acute kidney injury. These findings suggest that obinutuzumab holds great promise for use in non-tumor indications.

### 4.8. Limitation

Both the FAERS and JADER databases are recognized as self-reporting systems that are accessible to health professionals and the general public, and they have inherent limitations. For example, the data sources of FAERS encompass consumers, medical professionals, and pharmaceutical companies. In contrast, JADER mainly derives from medical professionals and medical institutions. This variety may result in inconsistencies and incomplete information within the reports. Moreover, spontaneous reporting systems generally encounter problems such as underreporting, duplicate reporting, and incomplete information, which can impact the accuracy and reliability of the data. Owing to the lack of validation of causal relationships, signal detection methods (such as the reporting odds ratio ROR) can merely offer statistical associations rather than prove the causal link between drugs and AEs. Neither database can provide the number of real-world users of obinutuzumab, which is necessary to calculate the incidence of a specific type of AEs. Furthermore, as patients with hematological malignancies may concurrently suffer from some other non-tumor diseases, these non-tumor conditions can potentially influence the efficacy and safety of obinutuzumab. Consequently, when comparing the differences in AEs induced by obinutuzumab in tumor and non-tumor indications, it is also necessary to consider the clinical context of obinutuzumab. Combining clinical trials with epidemiological methods may offer a more accurate assessment of the safety risks associated with obinutuzumab.

### 5. Conclusion

In conclusion, this study performed signal mining on the actual AE signal data associated with obinutuzumab in the FAERS and JADER databases. It comprehensively evaluated the potential risks in clinical use and offered safety references for clinical practice. Moreover, these two databases cover populations in Europe, America, and Asia, facilitating a more comprehensive analysis of the drug risks of obinutuzumab. In this study, the risk signals of infectious diseases and hematological and lymphatic system diseases accounted for a relatively large proportion among the top 30 PTs. 19 AEs not documented in the drug instruction were identified in the FAERS database, and 15 in the JADER database. In addition, compared with non-tumor indications, the frequency of AEs related to infection and IRRs was higher and the signals were stronger in tumor indications, while there were no significant differences in other AEs. During the treatment with obinutuzumab, patients should undergo regular hematological examinations, infection monitoring, and adverse reaction monitoring. For instance, obinutuzumab may increase the risk of infection, so it is necessary to closely monitor the patient's immune status. Regular follow-ups help doctors adjust the treatment plan in a timely manner. Patients aged 65 and above or those with comorbidities may be at a higher risk during treatment, and special attention should be paid to their tolerance. In the case of severe adverse reactions, such as infusion-related reactions, treatment may have to be interrupted or the dosage adjusted. Therefore, the use of obinutuzumab should be determined through individualized decisions that take into account the patient's personal characteristics, disease status, and treatment goals. Prior to treatment, doctors must comprehensively evaluate the risks and benefits for the patient, devise a rational treatment plan, and closely monitor adverse reactions and therapeutic effects throughout the treatment process. However, it is necessary to consider the inherent limitations of the FAERS and JADER data, along with potential confounding factors and biases. Thus, the analysis results should be interpreted with prudence. These findings offer valuable evidence to guide further research and inform the clinical practice of obinutuzumab treatment.

## Supporting information

**S1 Table. Four methods formula and threshold value.** ROR, reporting odds ratio; PRR, proportional reporting ratio; CI, confidence interval; $\chi^2$, chi-squared; IC, information component; IC025, the lower bound of 95% CI; EBGM, empirical Bayesian geometric mean; EBGM05, the lower bound of 95% CI.
(DOCX)

**S1 File. Code for formula calculation.**
(DOCX)

**S2 File. Demo from FAERS database.**
(XLSX)

**S3 File. The adverse event information in the FAERS database.**
(XLSX)

**S4 File. Demo from JADER database.**
(XLSX)

**S5 File. The adverse event information in the JADER database.**
(XLSX)

## Author contributions

**Conceptualization:** Xiangpeng Li.

**Data curation:** Haonan Liang, Ping Leng, Zhanqi Cao.

**Writing – original draft:** Xiangpeng Li, Zhanqi Cao.

**Writing – review & editing:** Haonan Liang, Ping Leng, Zhanqi Cao.

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
