## [Decision Letter · Decision Letter 0]

1 Aug 2025

PONE-D-25-32339Safety assessment of obinutuzumab: Real-world adverse event analysis based on the FAERS and JADER databases from 2013 to 2025PLOS ONE

Dear Dr. Cao,

Thank you for submitting your manuscript to PLOS ONE. After careful consideration, we feel that it has merit but does not fully meet PLOS ONE’s publication criteria as it currently stands. Therefore, we invite you to submit a revised version of the manuscript that addresses the points raised during the review process.

We look forward to receiving your revised manuscript.

Kind regards,

Nestor Oliva Damaso

Academic Editor

PLOS ONE

Journal Requirements: 

3. We note that your Data Availability Statement is currently as follows: [All relevant data are within the manuscript and its Supporting Information files]

Reviewers' comments:

Reviewer's Responses to Questions

**Comments to the Author**

1. Is the manuscript technically sound, and do the data support the conclusions?

Reviewer #1: Yes

Reviewer #2: Partly

2. Has the statistical analysis been performed appropriately and rigorously? 

Reviewer #1: Yes

Reviewer #2: No

3. Have the authors made all data underlying the findings in their manuscript fully available?

Reviewer #1: Yes

Reviewer #2: No

4. Is the manuscript presented in an intelligible fashion and written in standard English?

Reviewer #1: Yes

Reviewer #2: No

5. Review Comments to the Author

Reviewer #1: The authors have systematically and rigorously analyzed the adverse effects (AE)caused by obinutuzumab using two large (FAERS and JADER) databases.

The paper provides an interesting compilation of the drug's main AE. Considering the increasingly widespread use of this drug, it will be useful for the reader.

To enhance the interest of the paper, I suggest the following points:

1. AE may have a greater or lesser incidence and severity depending on the pathology being treated. This last aspect (indications) is poorly developed in the manuscript. They are only briefly discussed in the section about baseline characteristics, and the 5 main indications (CLL, follicular lymphoma, B-cell lymphoma, etc.) are presented in Table 3. However, these 5 top indications include “prescriptions for unknown causes” and "Other." It would be helpful to show the most common indications within the group “Other”.

2. Among the indications for obinutuzumab, its use in rheumatologic and systemic diseases will become increasingly important, given its greater efficacy than rituximab (e.g., lupus nephritis). It would be interesting to compare the frequency, characteristics, and severity of AE in hematologic malignancies (CLL, follicular or B-cell lymphomas, etc.) with non-tumoral indications.

3. In relation to the two previous points, the interest of this paper would be enhanced by analyzing AE of a recently published large trial of obinutuzumab in lupus nephritis (REGENCY; NEJM 2025; 392: 1471-83), using the same methodology applied in this paper and comparing them with the AE observed in other pathologies.

Reviewer #2: The authors conduct a large-scale, real-world pharmacovigilance analysis of adverse events (AEs) associated with obinutuzumab by analyzing the FAERS and JADER databases. They apply multiple signal detection algorithms (ROR, PRR, BCPNN, EBGM) to identify risk signals that are not currently documented in the product label. The study addresses a relevant clinical issue, post-marketing surveillance of monoclonal antibody therapies, and provides real-world evidence. However, the current manuscript lacks critical depth in interpretation, reproducibility, and clarity of English language, which limits its impact.

Main comments:

In my opinion, the study demonstrates insufficient depth in the interpretation of the data:

The authors mainly report signal values without providing a clinical context for many of the findings.

A more in-depth discussion is needed on how these adverse events may arise from a mechanistic point of view and how they compare with similar monoclonal antibodies (rituximab).

I recommend including a meta-analysis or comparative safety analysis with the existing literature to strengthen the conclusions, which appear to be only comments or opinions without scientific rigor.

Language and grammar require revision

The manuscript contains multiple grammatical and structural errors and variations that affect readability. A thorough review by a native English speaker is recommended.

Lack of methodological transparency

The search criteria and exact filters used for FAERS and JADER are not clearly described. Without this information, the analysis is not reproducible. I recommend including a flow chart showing the number of reports included/excluded.

Does not control for confounding or bias in spontaneous reports:

The study should better acknowledge and discuss the limitations of spontaneous reports, including under-reporting, duplicate reports, and lack of denominator data. Consider incorporating a refined risk signal strategy or validating signals with external databases.

Clarify statistical methods:

Formulas and thresholds are included, but justification for the choice of methods is lacking. Statistical corrections are also missing.

Weak conclusions and no meaningful clinical recommendations: The current conclusion merely repeats the results. Provide clinical recommendations (follow-up needs, high-risk patient profiles) and discuss implications for prescribing.

Consider moving the technical descriptions of the algorithms (i.e., BCPNN formulas) to the supplementary material to improve the flow of the manuscript.

Tables 4 and 5 are dense and difficult to interpret. Highlight their content more clearly.

Abbreviations (NHL, ... see text) should be defined consistently the first time they are used.

There are several repeated references to “not registered in the specification”; clearly define what “specification” refers to.

6. PLOS authors have the option to publish the peer review history of their article (what does this mean? ). If published, this will include your full peer review and any attached files.

**Do you want your identity to be public for this peer review?** For information about this choice, including consent withdrawal, please see our Privacy Policy .

Reviewer #1: **Yes: ** Manuel Praga

Reviewer #2: No

---

## [Author Response · Author response to Decision Letter 1]

1 Sep 2025

Respond to academic editor

Answer: Thank you very much for your valuable comments. We have revised the manuscript format as per the requirements.

Answer: The code has been uploaded as a supplementary information file (S1_File).

3. We note that your Data Availability Statement is currently as follows: [All relevant data are within the manuscript and its Supporting Information files]

Answer: The original data have been supplied as separate files within the Supporting Information files.

Respond to Reviewer 1

Reviewer #1: The authors have systematically and rigorously analyzed the adverse effects (AE)caused by obinutuzumab using two large (FAERS and JADER) databases.

The paper provides an interesting compilation of the drug's main AE. Considering the increasingly widespread use of this drug, it will be useful for the reader.

To enhance the interest of the paper, I suggest the following points:

1.AE may have a greater or lesser incidence and severity depending on the pathology being treated. This last aspect (indications) is poorly developed in the manuscript. They are only briefly discussed in the section about baseline characteristics, and the 5 main indications (CLL, follicular lymphoma, B-cell lymphoma, etc.) are presented in Table 3. However, these 5 top indications include “prescriptions for unknown causes” and "Other." It would be helpful to show the most common indications within the group “Other”.

Answer: Thank you very much for your valuable comments. We have raised the number of indications to 20.

2.Among the indications for obinutuzumab, its use in rheumatologic and systemic diseases will become increasingly important, given its greater efficacy than rituximab (e.g., lupus nephritis). It would be interesting to compare the frequency, characteristics, and severity of AE in hematologic malignancies (CLL, follicular or B-cell lymphomas, etc.) with non-tumoral indications.

Answer: As depicted in Figures 4 and 5, we supplemented the disparities in AEs induced by obinutuzumab in tumor and non-tumor indications.

3. In relation to the two previous points, the interest of this paper would be enhanced by analyzing AE of a recently published large trial of obinutuzumab in lupus nephritis (REGENCY; NEJM 2025; 392: 1471-83), using the same methodology applied in this paper and comparing them with the AE observed in other pathologies.

Answer: As the method employed in this article is not applicable to the large trial (REGENCY; NEJM 2025; 392: 1471-83), in Section 4.7, we have supplemented the relevant content regarding the efficacy and safety of obinutuzumab in lupus nephritis.

Reviewer #2: The authors conduct a large-scale, real-world pharmacovigilance analysis of adverse events (AEs) associated with obinutuzumab by analyzing the FAERS and JADER databases. They apply multiple signal detection algorithms (ROR, PRR, BCPNN, EBGM) to identify risk signals that are not currently documented in the product label. The study addresses a relevant clinical issue, post-marketing surveillance of monoclonal antibody therapies, and provides real-world evidence. However, the current manuscript lacks critical depth in interpretation, reproducibility, and clarity of English language, which limits its impact.

Main comments:

In my opinion, the study demonstrates insufficient depth in the interpretation of the data:

The authors mainly report signal values without providing a clinical context for many of the findings.

A more in-depth discussion is needed on how these adverse events may arise from a mechanistic point of view and how they compare with similar monoclonal antibodies (rituximab).

I recommend including a meta-analysis or comparative safety analysis with the existing literature to strengthen the conclusions, which appear to be only comments or opinions without scientific rigor.

Answer: Thank you very much for your valuable comments. In the discussion section, we explored the mechanisms underlying various AEs induced by obinutuzumab. Moreover, in sections 4.2 and 4.3, a meta-analysis was cited to compare the efficacy and safety differences between obinutuzumab and rituximab.

Language and grammar require revision

The manuscript contains multiple grammatical and structural errors and variations that affect readability. A thorough review by a native English speaker is recommended.

Answer: We have revised the language and grammar throughout the manuscript.

Lack of methodological transparency

The search criteria and exact filters used for FAERS and JADER are not clearly described. Without this information, the analysis is not reproducible. I recommend including a flow chart showing the number of reports included/excluded.

Answer: We have added a flow chart (Fig 1).

Does not control for confounding or bias in spontaneous reports:

The study should better acknowledge and discuss the limitations of spontaneous reports, including under-reporting, duplicate reports, and lack of denominator data. Consider incorporating a refined risk signal strategy or validating signals with external databases.

Answer: In section 4.8, we conducted a more in-depth analysis of the limitations of spontaneous reporting and data statistics.

Clarify statistical methods:

Formulas and thresholds are included, but justification for the choice of methods is lacking. Statistical corrections are also missing.

Answer: We have provided the rationale for the selection of methods in section 2.2. The reported statistics of adverse events from the FAERS and JADER databases are not appropriate for statistical corrections.

Weak conclusions and no meaningful clinical recommendations: The current conclusion merely repeats the results. Provide clinical recommendations (follow-up needs, high-risk patient profiles) and discuss implications for prescribing.

Answer: We have incorporated clinical recommendations and deliberated on the implications for prescribing within the conclusion.

Consider moving the technical descriptions of the algorithms (i.e., BCPNN formulas) to the supplementary material to improve the flow of the manuscript.

Answer: We have moved the technical descriptions of the algorithms to the supplementary material.

Tables 4 and 5 are dense and difficult to interpret. Highlight their content more clearly.

Answer: We have made revisions and improvements to Tables 4 and 5 to enhance comprehensibility.

Abbreviations (NHL, ... see text) should be defined consistently the first time they are used.

Answer: These problems have been rectified.

There are several repeated references to “not registered in the specification”; clearly define what “specification” refers to.

Answer: We have revised the “specification” to “drug instruction”.

---

## [Editor Report · Decision Letter 1]

25 Sep 2025

Safety assessment of obinutuzumab: Real-world adverse event analysis based on the FAERS and JADER databases from 2013 to 2025

PONE-D-25-32339R1

Dear Dr. Cao,

We’re pleased to inform you that your manuscript has been judged scientifically suitable for publication and will be formally accepted for publication once it meets all outstanding technical requirements.

Kind regards,

Nestor Oliva Damaso

Academic Editor

PLOS ONE

Additional Editor Comments (optional):

Thank you so much for your valuable study and for submitting to PloS One.

In my opinion this is an important paper helpful for clinicians that its interest will even rise with new indications of Obi in for example nephrology.

Best.

Nestor
---

## [Editor Report · Acceptance letter]

PONE-D-25-32339R1

PLOS ONE

Dear Dr. Cao,

I'm pleased to inform you that your manuscript has been deemed suitable for publication in PLOS ONE. Congratulations! Your manuscript is now being handed over to our production team.

Kind regards,

on behalf of

Dr. Nestor Oliva Damaso

Academic Editor

PLOS ONE